# Effects of Laser Treatment of Terbium-Doped Indium Oxide Thin Films and Transistors

**DOI:** 10.3390/nano14110908

**Published:** 2024-05-22

**Authors:** Rihui Yao, Dingrong Liu, Nanhong Chen, Honglong Ning, Guoping Su, Yuexin Yang, Dongxiang Luo, Xianzhe Liu, Haoyan Chen, Muyun Li, Junbiao Peng

**Affiliations:** 1Guangdong Basic Research Center of Excellence for Energy & Information Polymer Materials, State Key Laboratory of Luminescent Materials and Devices, School of Materials Sciences and Engineering, South China University of Technology, Guangzhou 510640, China; yaorihui@scut.edu.cn (R.Y.); 202220119085@scut.edu.cn (D.L.); chen-nanhong@foxmail.com (N.C.); 201730321254@mail.scut.edu.cn (G.S.); msyangyx@mail.scut.edu.cn (Y.Y.); 202030270082@mail.scut.edu.cn (H.C.); 202111084406@scut.edu.cn (M.L.); psjbpeng@scut.edu.cn (J.P.); 2Huangpu Hydrogen Innovation Center, Guangzhou Key Laboratory for Clean Energy and Materials, School of Chemistry and Chemical Engineering, Guangzhou University, Guangzhou 510006, China; 3Research Center of Flexible Sensing Materials and Devices, School of Applied Physics and Materials, Wuyi University, Jiangmen 529020, China; liuxianzhe@wyu.edu.cn

**Keywords:** laser processing, thin film transistor, terbium doping In_2_O_3_, bias light stability

## Abstract

In this study, a KrF excimer laser with a high-absorption coefficient in metal oxide films and a wavelength of 248 nm was selected for the post-processing of a film and metal oxide thin film transistor (MOTFT). Due to the poor negative bias illumination stress (NBIS) stability of indium gallium zinc oxide thin film transistor (IGZO-TFT) devices, terbium-doped Tb:In_2_O_3_ material was selected as the target of this study. The XPS test revealed the presence of both Tb^3+^ and Tb^4+^ ions in the Tb:In_2_O_3_ film. It was hypothesized that the peak of the laser thermal effect was reduced and the action time was prolonged by the f-f jump of Tb^3+^ ions and the C-T jump of Tb^4+^ ions during the laser treatment. Studies related to the treatment of Tb:In_2_O_3_ films with different laser energy densities have been carried out. It is shown that as the laser energy density increases, the film density increases, the thickness decreases, the carrier concentration increases, and the optical band gap widens. Terbium has a low electronegativity (1.1 eV) and a high Tb-O dissociation energy (707 kJ/mol), which brings about a large lattice distortion. The Tb:In_2_O_3_ films did not show significant crystallization even under laser energy density treatment of up to 250 mJ/cm^2^. Compared with pure In_2_O_3_-TFT, the doping of Tb ions effectively reduces the off-state current (1.16 × 10^−11^ A vs. 1.66 × 10^−12^ A), improves the switching current ratio (1.63 × 10^6^ vs. 1.34 × 10^7^) and improves the NBIS stability (ΔV_ON_ = −10.4 V vs. 6.4 V) and positive bias illumination stress (PBIS) stability (ΔV_ON_ = 8 V vs. 1.6 V).

## 1. Introduction

At present, new display technology products are endlessly emerging. People are increasingly demanding features of display devices, such as high resolution, thin, flexible, transparent and rich in color. The metal oxide thin film transistor (MOTFT) has the advantages of high mobility (1–100 cm^2^/V·s) and good film uniformity [1,2,3,4]. It has become a strong competitor in the display backplane industry represented by an active matrix liquid crystal display and active matrix organic light-emitting diode. 

The thin film transistor is a kind of field-effect transistor. TFT devices typically consist of an active layer, an insulating layer, a gate electrode, a source electrode and drain electrode. In TFTs, the most important material is the semiconductor active layer. According to the difference of semiconductor active layer materials, TFT can be divided into the following four categories: a-Si TFT, p-Si TFT, OTFT and MOS-TFT [5,6,7,8]. Among them, MOS-TFT has the advantages of high field-effect mobility, high uniformity, good electrical stability and high transparency, which are suitable for the future display preparation requirements, such as large size and flexibility [1,4].

The bottom of the conduction band of In_2_O_3_ material is formed by the hybridization of the 5s free electron state of In and the highly dispersed 2s electron state of O, while the valence band edge is formed by the hybridization of the 2P electron state of O and the 5d electron state of In. The unique band structure of In_2_O_3_ materials leads to the uniform distribution of carriers, which greatly reduces the scattering effect of carriers, so that In_2_O_3_ materials show higher carrier mobility [9]. The widely used IGZO materials are currently doped with Ga ions to suppress the oxygen vacancies in the material to reduce the carrier concentration. However, the bias light stability of IGZO-TFT devices is poor, resulting in the need to design complex compensation circuits to eliminate the drift of threshold voltage due to continuous light and negative gate bias in practical applications. The electronegativity of terbium (1.1 eV) is much lower than that of Ga materials, and the Tb-O dissociation energy (707 kJ/mol) is much higher than that of Ga-O. Therefore, only a small amount of terbium doping is required to effectively passivate the oxygen vacancies in oxide semiconductor materials and regulate the carrier concentration [10]. In addition, terbium also converts the white light incident to the material into a non-radiative leap through a charge-migration band jump between the Tb^4+^ ion and the ligand (C-T leap), improving the NBIS stability of the device [11].

The solution method has the advantages of being a simple process, having a high yield, a high material utilization and the easy control of chemical composition, which provides the possibility of preparing metal oxide semiconductors on a large scale. In studies of MOTFT solution preparation, the active layer consists mainly of precursors prepared by sol–gel method [12,13] or nanoparticles (NPs) dispersed in a carrier solvent [14,15,16], which are deposited on the substrate by spin-coating method, inkjet printing, etc. Whether the films are prepared by sol–gel or nanoparticle methods, post-processing is usually required after deposition to improve their properties [17].

The traditional heat treatment process has some disadvantages, such as high energy consumption, long treatment time, only providing the overall treatment of the device, high process temperature and the incompatibility of the flexible substrate [18]. As an emerging treatment technique in the field of flexible, printed and wearable devices, laser treatment effectively avoids the aforementioned drawbacks. Laser treatment can effectively treat precursor films or nanoparticle films by high-energy radiation and absorption of high-energy photons to convert them into metal oxide semiconductor films and improve their quality for practical applications. By adjusting the laser treatment parameters, such as laser intensity, pulse width and scan speed, the energy input to the film can be precisely controlled to achieve the desired thermal effect [19,20,21,22,23,24]. The heating and cooling rates of laser treatment (>10^6^ °C/s) are several orders of magnitude higher than conventional and rapid heat treatments, thus allowing for rapid treatment of thin films with minimal energy loss [25]. In addition, laser treatment is a top-down treatment technique that allows for precise control of the treatment location, so that the treatment area can be limited to a specific range in-plane and in the thickness direction, and can selectively process the trench area of the film and device to improve the performance of the film and MOTFT without affecting the substrate and adjacent materials [24,26,27,28].

## 2. Materials and Methods

In this study, a KrF excimer laser with a wavelength of 248 nm, which has a high absorption coefficient in metal oxide thin films, was selected for the post-processing of Tb:In_2_O_3_ thin films and TFT devices prepared by the solution method. In this work, we explore the effect of laser treatment of 248 nm KrF excimer on the performance of Tb:In_2_O_3_ thin films and Tb:In_2_O_3_-TFT devices, investigate the mechanism of laser-material interaction, and fabricate MOTFT devices with good stability.

The Tb:In_2_O_3_ thin film precursors were obtained by using indium nitrate hydrate as the indium source, terbium nitrate hydrate as the terbium source and ethylene glycol methyl ether as the solvent, and the ratio of indium to terbium In:Tb = 96:4 was added to ethylene glycol methyl ether at a concentration of 0.4 mol/L. The mixture was placed on a magnetic stirrer and stirred at room temperature for 12 h to obtain a stable and clarified Tb:In_2_O_3_ precursor solution.

The TFT device active layer precursor solution was prepared using indium nitrate hydrate as the indium source, terbium nitrate hydrate as the terbium source and 90% ethylene glycol + 10% deionized water as the solvent. The metal salts were added to the solvent at a ratio of 0.2 mol/L, and the ratio of indium to terbium was In:Tb = 96:4. The mixture was placed on a magnetic stirrer and stirred at room temperature for 12 h to obtain a stable and clarified precursor solution.

The TbO_X_ film precursor was made by using terbium nitrate hydrate as the terbium source and ethylene glycol methyl ether as the solvent, which was added to ethylene glycol methyl ether at a concentration of 0.4 mol/L. The mixture was placed on a magnetic stirrer and stirred at room temperature for 12 h to obtain a stable and clarified TbO_X_ precursor solution.

The 1 cm × 1 cm quartz glass substrates were cleaned with deionized water and isopropanol by ultrasonic cleaning for 15 min, followed by drying in an oven at 80 °C. The substrate was treated with UV for 20 min to improve the hydrophilicity of the substrate before rotary coating. The precursor solution of Tb:In_2_O_3_ thin film (In:Tb = 96:4, 0.4 mol/L) and TbO_x_ film (0.4 mol/L) was filtered with an organic phase filter tip of 0.45 μm, and 25 μL of the filtered precursor solution was added dropwise onto the substrate using a pipette gun, homogenized at 500 rpm for 6 s, and shaken off at 8000 rpm for 20 s. The wet film after spin coating was dried at low temperature for 10 min on a hot table at 40 °C to obtain a homogeneous active layer film, and then warmed up with the oven to a temperature rise of 150 °C for 15 min. The Tb:In_2_O_3_ and TbO_X_ films were produced by pretreatment at 150 °C for 15 min to further remove the residual solvent in the active layer films and to promote the sol–gel process.

An approximately 300 nm thick conductive film of Al:Nd was deposited on a washed glass substrate using DC sputtering and a wet etching process to form a T-shaped pattern. The patterned Al:Nd conductive film was oxidized using an anodic oxidation process to form an approximately 200 nm thick AlO_x_:Nd insulating layer, and the unoxidized Al:Nd conductive film was used as the gate electrode. The substrate was cleaned by ultrasonic cleaning with deionized water and isopropyl alcohol for 15 min each, and then dried in an oven at 80 °C. In order to achieve better device performance, the Tb:In_2_O_3_ active layer in this study was patterned by UV light treatment. The substrate was placed on a mask plate with the patterned active layer, and after 20 min of UV light treatment, a hydrophilic surface was formed in the active layer region, while the non-UV irradiated region remained hydrophobic. The precursor solution of the TFT device active layer (In:Tb = 96:4, 0.2 mol/L) was filtered using a 0.45 μm organic phase filter, and 25 μL of the filtered precursor solution was added dropwise onto the UV-patterned substrate using a pipette gun, followed by homogenization at 500 rpm for 6 s and shaking at 8000 rpm for 20 s. The spin-coated wet film was dried at low temperature for 10 min under a hot table at 40 °C to obtain a uniform Tb:In_2_O_3_ active layer film. The film was then pre-treated at 150 °C for 15 min to further remove the residual solvent in the Tb:In_2_O_3_ active layer film and to promote the sol–gel process (Figure 1).

The Tb:In_2_O_3_ films were treated with 20 pulses and laser energy densities of 130, 160, 190, 220 and 250 mJ/cm^2^. The TbO_X_ films were treated with 20 pulses and laser energy densities of 190 mJ/cm^2^. the Tb:In_2_O_3_-TFT devices were treated with 20 pulses and laser energy densities of 105, 115, 125, 135, 145 and 155 mJ/cm^2^. 

Characterization of the lattice structure of the film was by X-ray diffraction (XRD, PANalytical 7602 EA, Almelo, The Netherlands). The thickness and density of the films are characterized by X-ray reflection (XRR, PANalytical 7602 EA). The surface morphology of the films was determined by atomic force microscopy (AFM, BY 3000, Being Nano-Instruments, Beijing, China). Solvent residues of the films was determined by Fourier-transform infrared spectroscopy (FTIR, Shimadzu IRPrestige21, Kyoto, Japan). Characterization of electrical properties of films was determined by Hall testing (ECOPIA HMS 5300, Pyeongchang-gun, Republic of Korea). Characterization of the optical properties of the films was determined by UV–optical spectrophotometry (UV-2600, Shimadzu). Characterization of the photoluminescence spectra of thin films was determined by using fluorescence spectroscopy (PL, FluoroMax, Horiba, Kyoto, Japan). Characterization of the bonding state of the films was determined by X-ray photoelectron spectroscopy (XPS, ThermoFisher Nexsa, Waltham, MA, USA). The electrical properties of devices were determined by semiconductor parameter analyzer (FS-Pro, Primarius Technologies, Shanghai, China).

## 3. Results and Discussion

Figure 2 shows the UV–Vis Absorbance spectrum of the precursor solution of TbO_X_ (0.4 mol/L). It can be seen that there is a sharp linear peak near the wavelengths of 248 nm and 287 nm, which is typical of the Absorbance peak of the f-f leap in terbium ion. Therefore, it is speculated that the terbium ion in the film can also absorb photon energy and undergo f-f leap when the Tb:In_2_O_3_ film was treated with a KrF excimer laser at a 248 nm wavelength.

To verify the valence state of terbium ions in Tb:In_2_O_3_ films, XPS tests were performed. The XPS spectra of the 3d orbitals of Tb ions reflect the jump from the 3d to the 4f energy level, which are 3d^10^f^8^(^7^F_6_) → 3d^9^4f^9^ and 3d^10^f^7^(^8^S_12_) → 3d^9^4f^8^ for the Tb^3+^ and Tb^4+^ ions, respectively, with the corresponding 3d_5/2_ orbitals at 1239.1 eV and 1241.4 eV, respectively. The XPS spectra of the two valence ions overlap when they are present at the same time [29]. Figure 3 shows the XPS refined spectra of the Tb 3d_5/2_ orbitals of the Tb:In_2_O_3_ films and their fitting results, where it can be seen that some amount of Tb^3+^ ions and Tb^4+^ ions are present in all films.

Figure 4 shows the absorption spectra of TbO_X_ films in this study and the absorption spectra of Tb^3+^ ions and Tb^4+^ ions reported in the literature [30]. It can be seen that the broad-spectrum absorption caused by the C-T jump between the 4-valent Tb^4+^ ion and the ligand (oxygen) was present in the TbO_X_ film [11,31]. In TbO_X_ and Tb:In_2_O_3_ materials, the valence band is mainly composed of O 2p orbitals. During laser treatment of TbO_X_ films, the films absorbed photon energy and caused the electrons in the ligand orbitals to be excited to form electron–hole pairs, and the excited electrons migrate to the 4f orbitals of Tb^4+^ ions and reduce them to the excited state of (Tb^3+^)*. The process is shown in the energy band diagram as electron excitation from the valence band to Tb^3+^, as shown in Figure 5. In the laser treatment of Tb:In_2_O_3_ films, the Tb^4+^ ion can be reduced to the excited state by accepting electrons generated by the excitation of the oxygen vacancy (O_V_) in the indium oxide matrix as the donor, in addition to the electrons excited by its surrounding oxygen atoms. Since the energy difference between O_V_ and Tb^3+^ is smaller than that between valence band and Tb^3+^, the C-T leap between Tb^4+^ and O_V_ is more likely to occur, as shown in Figure 5. After the Tb^4+^ ion was excited and reduced to the (Tb^3+^)* excited state, electrons were again released and oxidized back to Tb^4+^ in the form of radiative or non-radiative leaps. The overall process is shown in Equations (1) and (2):Tb^4+^ + O^2−^ + Photon → (Tb^3+^)* + O^1−^ → Tb^4+^ + O^2−^ + Energy(1)
Tb^4+^ + O_V_ + Photon → (Tb^3+^)* + O_V_^+^ → Tb^4+^ + O_V_ + Energy(2)

After the excitation of terbium to the excited state, it usually returns to the ground state in the form of radiative or non-radiative leap. In order to investigate the transition of terbium back to the ground state after excitation with an excimer laser at 248 nm, PL tests were performed on quartz substrates, TbO_X_ films and Tb:In_2_O_3_ films. The excitation wavelength used was 248 nm, which corresponds to the wavelength of the KrF laser used in the experiment, and the test results are shown in Figure 6. It can be seen that the luminescence peaks of the quartz substrate appear at 500 nm and 750 nm, among which the peak at 500 nm is distorted due to the luminescence intensity exceeding the range of the instrument. In contrast to the luminescence spectrum of the substrate, it can be seen that the TbO_X_ and Tb:In_2_O_3_ films only show the luminescence peaks formed by the luminescence of the substrate under the effect of excitation light at 248 nm, and the luminescence peaks at 500 nm and 750 nm are not typical of terbium ions. Therefore, it can be speculated that the terbium in the Tb:In_2_O_3_ films returns to the ground state in the form of non-radiative leap after excitation.

Synthesizing the above experimental results, it can be speculated that during the laser treatment of Tb:In_2_O_3_ films, terbium ions play the following roles: (1) they directly absorb laser photon energy and undergo an f-f leap to the excited state; (2) the ligand oxygen atoms and oxygen vacancies in the indium oxide matrix absorb laser photon energy to generate electron–hole pairs, and electrons migrate to the 4f orbitals of Tb^4+^ ions and reduce them to the excited state of (Tb^3+^)*, which undergoes a C-T leap; (3) the excited state of terbium ion releases energy in the form of a non-radiative leap, which transfers energy to the lattice and triggers lattice vibrations.

In the laser treatment of pure In_2_O_3_ films, all the laser energy is transferred directly to the lattice, while in the laser treatment of Tb:In_2_O_3_ films, most of the energy is transferred to the indium oxide matrix and a small fraction of the energy will act on the terbium ion. The terbium ion absorbs the laser energy and is excited to the excited state, which then transfers the energy to the lattice in the form of a non-radiative leap. In the presence of terbium ions, the peak of the laser effect is reduced but the action time is prolonged, as shown in Figure 7.

Figure 8 shows the XRD pattern of Tb:In_2_O_3_ thin film at different laser energy densities. It can be seen that the films treated at different laser energy densities all have only a typical amorphous diffuse reflection peak and no sharp and obvious crystalline peak is observed. This indicates that the films generally maintain a good amorphous structure even at laser energies up to 250 mJ/cm^2^, which is favorable for their application in large-scale device arrays. In this respect, this study suggests that the amorphous structure is caused by the doping of terbium elements. On the one hand, the dissociation energy of the Tb-O (707 kJ/mol) bond is much higher than that of the In-O (346 kJ/mol) bond, making a higher energy required to break the Tb-O bond and undergo atomic rearrangement. On the other hand, the ionic radius of Tb^3+^ ions (0.092 nm) is much larger than that of In^3+^ ions (0.08 nm), which leads to a large lattice distortion in the thin film material and allows the film to maintain an amorphous structure even under higher laser energy density treatment. In addition, the peak laser effect is also weakened in the presence of terbium ions, and the laser energy density threshold required for film crystallization is increased.

Figure 9 shows the AFM test results of the Tb:In_2_O_3_ films treated with different laser energy densities. It can be seen that the roughness of the untreated film is 3.05 nm. After laser treatment, the film roughness is all below that of the untreated one. With the effect of terbium ions, the duration of laser action is prolonged and the atoms have more sufficient time to undergo rearrangement, resulting in a flatter surface and dense structure. The lowest film roughness of 1.12 nm was obtained at a laser energy density of 130 mJ/cm^2^, and the film roughness increased with increasing laser energy density. The roughness increased sharply to 2.97 nm when the laser energy density was increased to 250 mJ/cm^2^, but also remained at a low level. Since the laser treatment is performed in a top-down process, a weak crystallization may have occurred on the film surface at the laser energy density of 250 mJ/cm^2^, leading to the increase in roughness. This result indicates an improvement in the quality of the laser-treated films. 

Figure 10 shows the FTIR test results of the Tb:In_2_O_3_ films treated with different laser energy densities. It can be seen that the test curves of the films have a distinct absorption peak in the wavenumber range of 1000 cm^−1^–1400 cm^−1^. This peak is considered to be the superposition of the absorption peak of fatty ether in the wave number range of 1300 cm^−1^–1000 cm^−1^ and the absorption peak of nitrate in the wave number range of 1450 cm^−1^–1300 cm^−1^. The nitrate absorption peak is at 1540 cm^−1^. It can be seen that the peak signal is stronger for the untreated film, and the transmission of the absorption peak at 1000 cm^−1^–1400 cm^−1^ is as low as 88.2%, with more solvent molecules and nitrate remaining in the film. After 20 pulses of laser treatment, the residual solvent molecules and nitrate ions were significantly removed by the photoactivation and laser thermal effects, and the absorption peak transmittance at 1000 cm^−1^–1400 cm^−1^ increased to over 93.9%. The transmittance also increased gradually with increasing laser energy density, reaching 96.55% at the laser energy density of 250 mJ/cm^2^. This indicates that the laser treatment effectively removed the residual solvent molecules and nitrate ions from the film. The reduction in impurities reduces the defect density of the film and improves the orderliness of the lattice structure, which helps to enhance the electrical properties of the film.

Figure 11a shows the XRR test curves of the Tb:In_2_O_3_ films treated with different laser energy densities. The test curves were fitted by X’Pert Reflectivity software to obtain the thickness and density information of the films, as shown in Figure 11b. It can be seen that the untreated films have a lower density and a higher thickness due to the presence of more impurities and the lack of condensation reaction to form the metal oxide structure. Compared to the untreated sample, the density of the films treated with 20 pulses and 130 mJ/cm^2^ laser increased (3.53 vs. 3.5952 g/cm^3^) and the thickness decreased (50.55 vs. 46.43 nm). The film thickness continued to decrease and the density increased with increasing laser energy density. At the laser energy density of 250 mJ/cm^2^, the thickness decreased to 41.13 nm and the density increased to 3.8 g/cm^3^. This can be attributed to the removal of impurities and densification phenomenon within the film by the laser treatment, as shown by the FTIR test results, which significantly removed the residual solvent molecules and nitrate ions within the film, thus removing impurities. High-energy photons acting on the film produce atomic bond breaking and atomic rearrangement and thermal effects, atoms undergo relaxation rearrangement, non-dense structures such as voids in the film material are eliminated, atoms are tightly packed and the density of the film is increased. As the laser energy density increases, the interaction between the laser and the film becomes more significant, the depth of the action becomes deeper, the treatment of the film becomes more thorough, the density further increases and the thickness decreases.

Figure 12a–e shows the full spectrum of the XPS test of Tb:In_2_O_3_ thin film, and Figure 12f shows the percentage of the area occupied by C1s peaks. It can be seen that after 20 pulses and 130 mJ/cm^2^ laser treatment, the area occupied by the C1s peak of the film is lower and fewer impurities remain. As the laser energy density increases, the percentage of the area occupied by C1s peaks decreases and the impurities are removed more completely. The FTIR test results also support this conclusion. 

Figure 13a–e shows the refined spectrum of the O1s peak and the results of split peak fitting, the low-binding energy peak corresponds to the lattice oxygen (O_L_) in the film, the medium binding energy peak corresponds to the oxygen vacancy (O_V_) in the film, and the high-binding energy peak corresponds to the impurity oxygen (O_A_) in the film. Figure 12f shows the relative ratio of O_L_/O_A_ and O_V_ content of the film. According to Spilios et al., the degree of precursor (In:Tb = 96:4, 0.4 mol/L) conversion to metal oxides can be assessed by the ratio of O_L_/O_A_ [31]. It can be seen that the relative ratio of O_L_/O_A_ of the film reached 1.54 for a laser energy density of 130 mJ/cm^2^ and 20 laser pulse treatment, and the transition of the film to the metal oxide state is more complete. As the laser energy density increases, the relative O_L_/O_A_ ratio further increases and the degree of lattice ordering increases, reaching 3.51 at the laser energy density of 250 mJ/cm^2^.

Tb:In_2_O_3_ films treated with different laser energy densities were tested with a Hall effect tester to characterize their electrical properties. The data for the untreated Tb:In_2_O_3_ films are not presented here because the resistivity was too high and beyond the reach of the instrument. Figure 14 shows the change curve of the electrical properties of the Tb:In_2_O_3_ films after the laser treatment. It can be seen that the film carrier concentration rises with increasing laser energy density and increases sharply at a laser energy density of 250 mJ/cm^2^ (−3.65 × 10^19^ cm^−3^). The Hall mobility of the film increases and then decreases with increasing laser energy density, reaching a maximum value of 0.838 cm^2^/V·s at 190 mJ/cm^2^ and a minimum value of 0.261 cm^2^/V·s at 250 mJ/cm^2^. In metal oxide semiconductors, the electrical properties follow the permeation conductivity model [32]. When the carrier concentration is low, its carriers choose to avoid higher energy barriers and transport through longer transport paths, when the mobility of the film is low. As the carrier concentration increases, the Fermi energy level rises and the relative energy difference between it and the energy barrier decreases, allowing the carriers to be transported through shorter paths or across the energy barrier, and the film mobility is enhanced. With the further increase in laser energy density, combined with the XPS test results, it can be seen that the oxygen vacancy concentration as the sender energy level further increases and the carrier concentration continues to rise, but a too-high defect state concentration and too-high carrier concentration lead to carrier transport by ionized impurity scattering and carrier scattering, and the mobility decreases.

The optical properties of the Tb:In_2_O_3_ films were characterized by UV–Vis spectrophotometer, and the transmittance curves are shown in Figure 15a. The optical band gap of the films was fitted by Tacu’s equation, and the fitting results are shown in Figure 15b. From the transmittance curves, it can be seen that the laser treatment densifies the film and reduces the surface roughness and light scattering compared to the untreated Tb:In_2_O_3_ film. The transmittance of the laser-treated films was increased in the 450 nm-800 nm band, reaching more than 97.5%. In addition, the absorption edge of the transmittance curve of the laser-treated Tb:In_2_O_3_ films showed a blueshift with increasing laser energy density, and the fitted optical band gap curves also showed a broadening of the optical band gap of the film with increasing laser energy. The blue shift of the absorption edge and the broadening of the optical bandgap at the laser energy density of 250 mJ/cm^2^ were found to be caused by the change in the carrier concentration compared to the Hall test results. According to the Burstein–Moss effect, the increase in carrier concentration leads to the Fermi energy level of the material entering the conduction band and the electron filling to higher energy levels above the conduction band, resulting in an increase in the energy required for the electron leap, which corresponds to the widening of the optical band gap.

Figure 16 shows the AFM images of the Tb:In_2_O_3_ active layer films of Tb:In_2_O_3_-TFT devices treated with different laser energy densities. It can be seen that the laser-treated active layer films all have a roughness below 0.7 nm and a flat surface, which is favorable for the deposition of Al electrodes and the formation of good ohmic contacts.

Figure 17 shows the output curves of the Tb:In_2_O_3_-TFT treated with different laser energy densities. It can be seen that at the laser energy density of 105 mJ/cm^2^, there is no significant output curve even when a gate voltage of 20 V is applied to the device. Except for the laser energy density of 105 mJ/cm^2^, all devices exhibit obvious characteristics of n-type thin film transistors. At a laser energy density of 115 mJ/cm^2^, the device is switched on only when the gate voltage is applied to 20 V, and the current decreases rapidly with increasing drain-source voltage after the switch on, resulting in poor device performance. At a laser energy density of 125 mJ/cm^2^, a more obvious current pinch-off effect occurs when a gate voltage of 15 V is applied. However, when the gate voltage is applied to 20 V, the current decreases rapidly with increasing drain-source voltage after turning on again. As laser energy density is further increased, the devices treated with 135 mJ/cm^2^ and 145 mJ/cm^2^ started to show an obvious current pinch-off effect without current crowding, indicating a good ohmic contact between the Al electrode and the Tb:In_2_O_3_ active layer film. When the laser energy density was increased to 155 mJ/cm^2^, the device current pinch-off effect started to disappear and the current was no longer saturated under this measurement condition.

Figure 18 shows the In_2_O_3_-TFT transfer curves for different laser energy density treatments, from which the electrical performance parameters of the TFT devices are extracted as shown in Table 1.

Figure 19 shows the Tb:In_2_O_3_-TFT transfer curves for different laser energy density treatments, from which the electrical performance parameters of the TFT devices are extracted as shown in Table 2. 

It can be seen that the device does not exhibit a significant switching characteristic curve at the laser energy density of 105 mJ/cm^2^, and its source drain current decreases with increasing gate voltage. At 115 mJ/cm^2^, the device starts to show switching properties, and the off-state current is as low as 10^−11^ A. However, the instrumental measurements fluctuate greatly at this current level, and the off-state current curve fluctuates greatly due to serious interference from the external environment. The off-state current and device switching current ratios extracted at this time are not credible. Combined with the output curve, it can also be seen that the device performance of the 115 mJ/cm^2^ treatment is poor, and its mobility is as low as 0.022 cm^2^/V·s. As the laser energy density increases further, the device active layer carrier concentration increases, the threshold voltage gradually decreases, the on-state current increases, the subthreshold swing decreases and the device mobility gradually increases. At 145 mJ/cm^2^, the device switching current ratio reaches a maximum of 1.34 × 10^7^, at which the mobility increases to 1.07 cm^2^/V·s and the subthreshold swing is as low as 0.43 V/decade. As can be seen, the performance of the laser-treated Tb:In_2_O_3_-TFT device is slightly better than that of the heat-treated device.

Figure 20 shows the pattern of variation of the transfer characteristic curves over time during the NBIS test for the pure In_2_O_3_-TFT device and the Tb:In_2_O_3_-TFT device. With a gate bias of −20 V applied for 3600 s and a light intensity of 250 Lux, the turn-on voltage of the pure In_2_O_3_-TFT device shifted from −2.4 V to −12.8 V with ΔV_ON_ = −10.4 V, while the turn-on voltage of the Tb:In_2_O_3_-TFT device shifted from 0.2 V to −6.6 V with ΔV_ON_ = −6.4 V. It can be seen that, compared with the pure device, it can be seen that the NBIS stability of the Terbium-doped Tb:In_2_O_3_-TFT device is significantly improved compared to the pure In_2_O_3_-TFT device (ΔV_ON_ = −10.4 V vs. ΔV_ON_ = −6.4 V). Figure 21 illustrates the transfer characteristic curves with time during PBIS testing for pure In_2_O_3_-TFT devices and Tb:In_2_O_3_-TFT devices. With a gate bias of 20 V applied for 3600 s and a light intensity of 250 Lux, the turn-on voltage of the pure In_2_O_3_-TFT device shifted from −2.4 V to 5.6 V with ΔV_ON_ = 8 V, while the turn-on voltage of the Tb:In_2_O_3_-TFT device shifted from 0.2 V to 1.8 V with ΔV_ON_ = 1.6 V. Compared with the pure In_2_O_3_-TFT, it can be seen that the PBIS stability of the Tb:In_2_O_3_-TFT device doped with terbium is significantly more improved than the pure In_2_O_3_-TFT device (ΔV_ON_ = 8 V vs. ΔV_ON_ = 1.6 V).

For the enhancement of the device stability after doping, this study suggests that there may be two possible reasons: (1) the passivation of oxygen vacancies by terbium element and (2) the down-conversion effect of terbium ion [12]. It has been commonly suggested that neutral oxygen vacancies lose electrons to form O_V_^2+^ upon incident light irradiation, which in turn leads to a drift in turn-on voltage. The doped terbium element has a low electronegativity (1.1 eV) and a high Tb-O dissociation energy (707 kJ/mol), which can effectively passivate the oxygen vacancies and thus improve the device stability. In addition, terbium can be converted to non-radiative leap by C-T leap under incident white light, which further reduces the probability of the oxygen vacancies being excited by incident light and improves the device stability.

A comparison of Tb:In_2_O_3_-TFT devices with In_2_O_3_-TFT devices shows that the doping of Tb ions effectively improves the switching current ratio and NBIS stability of TFT devices by passivating oxygen vacancies and converting to non-radiative leaps for incident light [12], but reduces the device mobility. Compared with the pure In_2_O_3_-TFT device, the open-state currents of both are at the same level (1.88 × 10^−5^ A vs. 2.23 × 10^−5^ A). However, due to the passivation of oxygen vacancies by Tb ions, the off-state current of the Tb:In_2_O_3_-TFT device is one order of magnitude lower than that of the pure In_2_O_3_-TFT device (1.16 × 10^−11^ A vs. 1.66 × 10^−12^ A), effectively improving the switching current ratio. However, the doping of Tb ions also has a certain impact on the lattice orderliness, resulting in a lower mobility of the doped device (1.31 cm^2^/V·s vs 1.07 cm^2^/V·s). In addition, from the treatment process, the pure In_2_O_3_-TFT devices already exhibit certain switching characteristics and output curves at a laser energy density of 105 mJ/cm^2^, and the device performance is optimized when increasing to 125 mJ/cm^2^. At a laser energy density of 145 mJ/cm^2^, the device performance has deteriorated significantly, with an increase in the output current fluctuations and a sharp increase in subthreshold swing. In contrast, the Tb:In_2_O_3_-TFT device has no switching characteristics at a laser energy density of 105 mJ/cm^2^, and the device performance is optimized when the laser energy density is increased to 145 mJ/cm^2^, and still shows good device performance at up to 155 mJ/cm^2^. In this study, it is concluded that the doped terbium reduces the instantaneous peak temperature of laser treatment and prolongs the laser duration through the transition process of ground state → excited state → ground state on the one hand. On the other hand, the high Tb-O bonding energy of terbium also increases the energy threshold required for laser treatment, which makes the Tb:In_2_O_3_-TFT devices require a higher laser energy density treatment than the In_2_O_3_-TFT devices to show good device performance.

## 4. Conclusions

In this study, the interaction between terbium element and laser was investigated. During the laser treatment of Tb:In_2_O_3_ films, terbium ions play the following roles: (1) they directly absorb laser photon energy and undergo an f-f leap to the excited state; (2) the ligand oxygen atoms and oxygen vacancies in the indium oxide matrix absorb laser photon energy to generate electron–hole pairs, and electrons migrate to the 4f orbitals of Tb^4+^ ions and reduce them to the excited state of (Tb^3+^)*, undergoing a C-T jump; (3) the excited state of the terbium ion releases energy in the form of a non-radiative jump, which transfers energy to the lattice and triggers lattice vibrations. In the presence of terbium ions, the peak of the laser effect decreases, but the action time is prolonged.

In this study, the effect of 20 laser pulses with different laser energy density treatments on the Tb:In_2_O_3_ films was investigated. Impurities such as residual solvents and metal ligands were significantly removed from the laser-treated films, and the conversion from precursors (In:Tb = 96:4, 0.4 mol/L) to metal oxides was completed by condensation reactions. The non-dense structures such as voids in the film material are eliminated, the atoms are tightly packed, the film density increases, the thickness decreases and the surface roughness decreases. Under the effect of the terbium element, the films maintain a good amorphous structure even with a high laser energy density treatment. The electrical properties of the films were also improved after laser treatment, with an increase in the film carrier concentration and an increase in the Hall mobility. The films maintained good transmittance before and after laser treatment, and the transmittance absorption edge was blue-shifted and the optical band gap was widened with increasing laser energy density. 

The laser-treated Tb:In_2_O_3_-TFT active layer films exhibit low roughness, which facilitates the formation of ohmic contacts with Al electrodes. The laser-treated Tb:In_2_O_3_-TFT devices exhibit good switching characteristics and device performance. As the laser energy density increases, the device saturation mobility and the switching current ratio all increase and the threshold voltage decreases. The best performance is achieved at the laser energy density of 145 mJ/cm^2^ with a saturation mobility of 1.07 cm^2^/V⸱s, a switching current ratio of 1.34 × 10^7^ and a subthreshold swing as low as 0.43 V/decade. Compared with pure In_2_O_3_-TFT, the doping of Tb ions effectively improves the NBIS stability (ΔV_ON_ = −10.4 V vs. ΔV_ON_ = −6.4 V) and PBIS stability (ΔV_ON_ = 8 V vs. ΔV_ON_ = 1.6 V) of TFT devices. This study presents a new idea for indium-based TFT element doping: terbium doping, and helps to raise people’s attention to the effect of laser treatment on the optimization of thin film properties.

## Figures and Tables

**Figure 1 nanomaterials-14-00908-f001:**
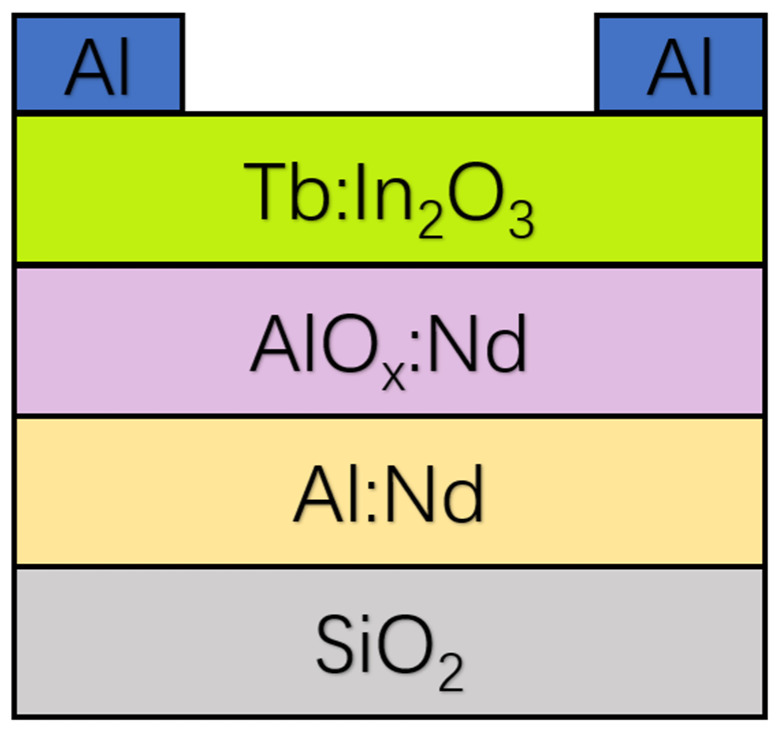
Schematic diagram of TFT device structure. (Thickness information: Al:Nd = 100 nm, AlO_x_:Nd = 200 nm, Tb:In_2_O_3_ = 40 nm, Al = 115 nm.)

**Figure 2 nanomaterials-14-00908-f002:**
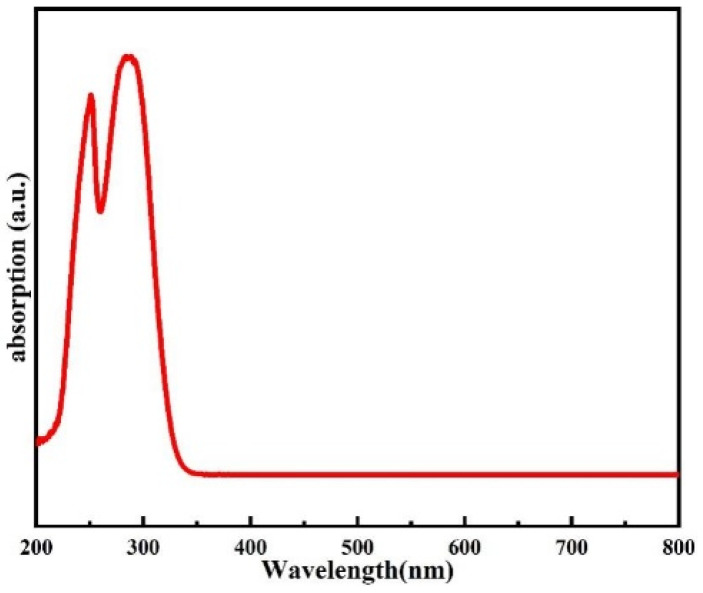
Absorbance spectrum of precursor solution of TbO_X_ (0.4 mol/L).

**Figure 3 nanomaterials-14-00908-f003:**
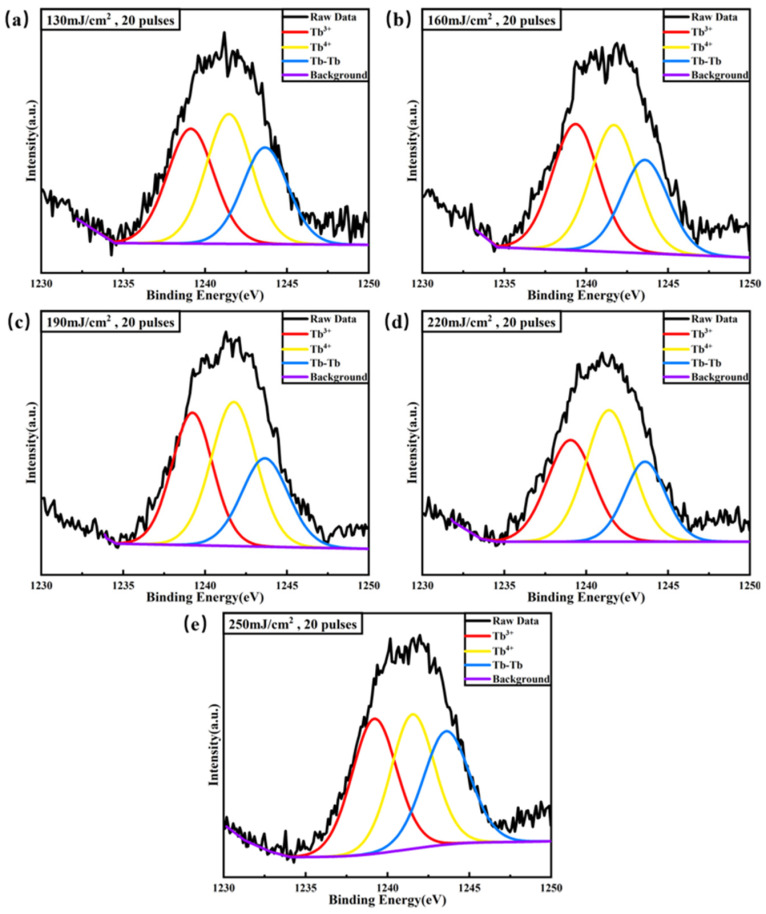
Tb 3d_5/2_ refined spectrum and fitting results of Tb:In_2_O_3_ thin film treated with different energy densities. (**a**) 130 mJ/cm^2^; (**b**) 160 mJ/cm^2^; (**c**) 190 mJ/cm^2^; (**d**) 220 mJ/cm^2^; (**e**) 250 mJ/cm^2^.

**Figure 4 nanomaterials-14-00908-f004:**
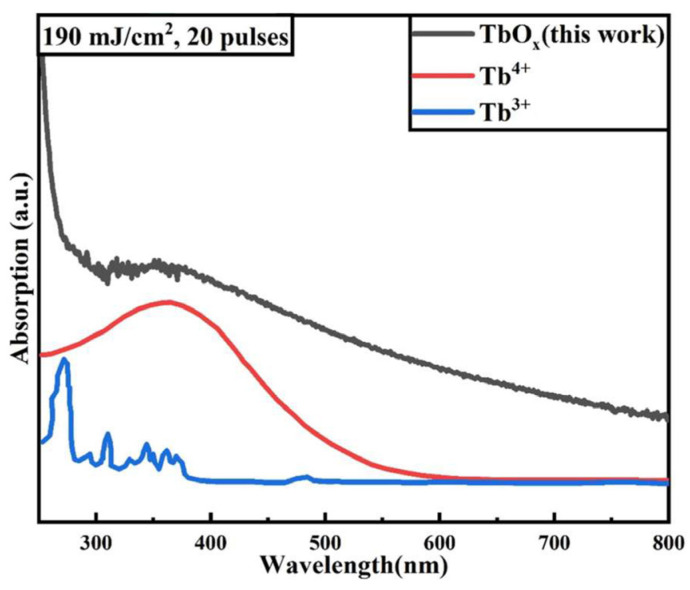
TbO_X_ thin film absorption spectra and reported Tb^3+^ and Tb^4+^ion absorption spectra in the literature.

**Figure 5 nanomaterials-14-00908-f005:**
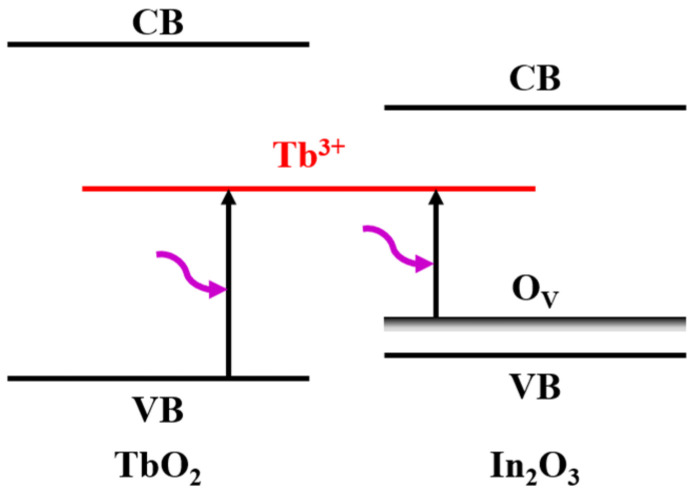
Schematic diagram of C-T transition between Tb^4+^ ion and ligand oxygen/oxygen vacancy.

**Figure 6 nanomaterials-14-00908-f006:**
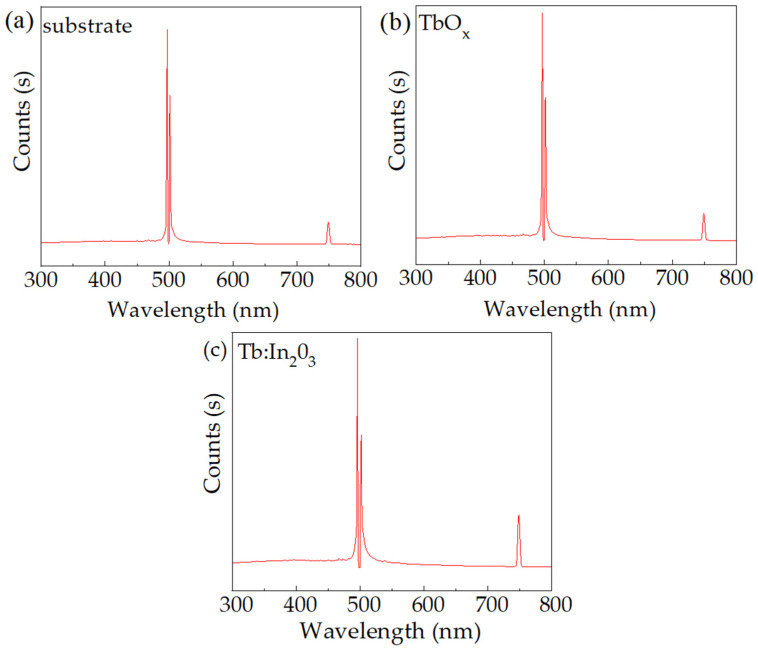
Fluorescence spectrum test results: (**a**) Quartz substrate; (**b**) TbO_X_ thin film; (**c**) Tb:In_2_O_3_ thin film.

**Figure 7 nanomaterials-14-00908-f007:**
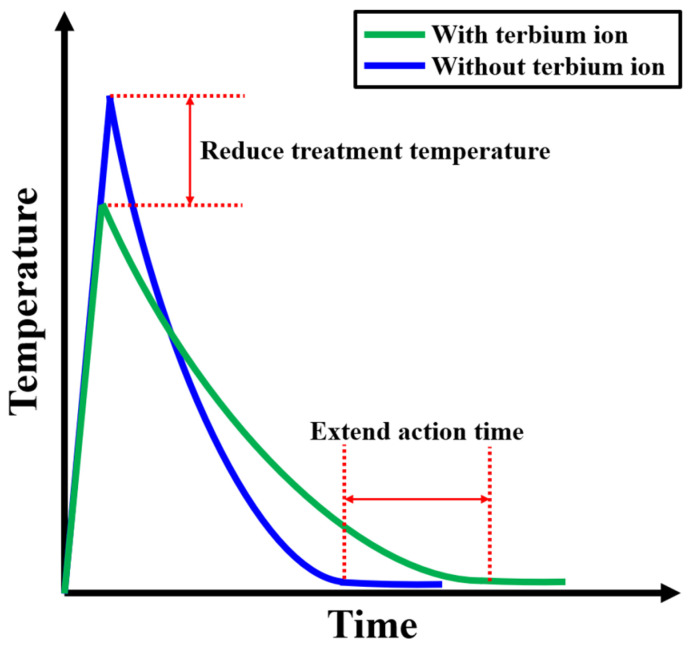
Theoretical diagram of the effect of Tb ions on thermal effect of laser treatment.

**Figure 8 nanomaterials-14-00908-f008:**
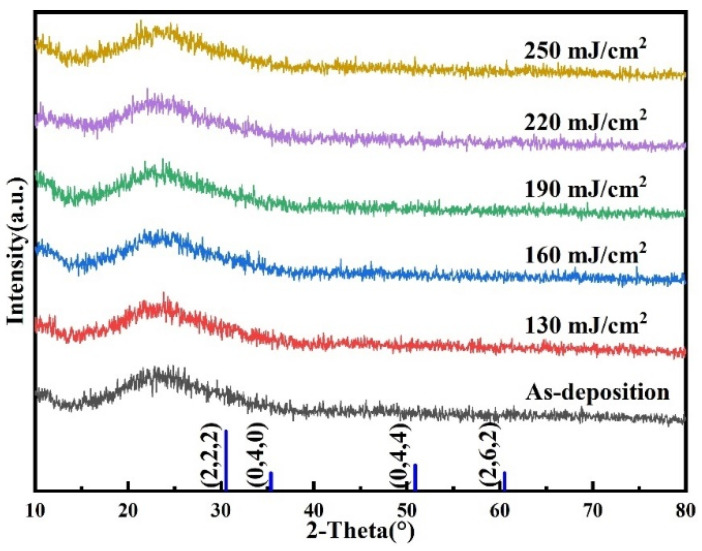
XRD diagram of Tb:In_2_O_3_ thin films at different laser energy densities.

**Figure 9 nanomaterials-14-00908-f009:**
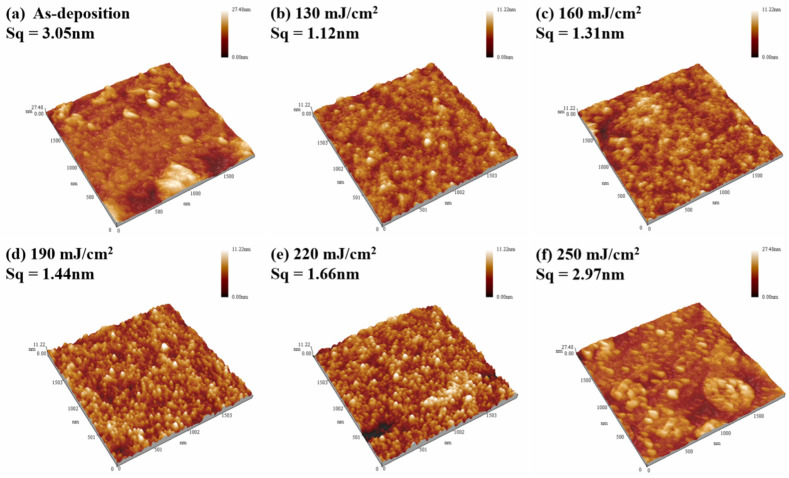
Surface topography of thin films treated with different energy densities. (**a**) As-deposition; (**b**) 130 mJ/cm^2^; (**c**) 160 mJ/cm^2^; (**d**) 190 mJ/cm^2^; (**e**) 220 mJ/cm^2^ and (**f**) 250 mJ/cm^2^.

**Figure 10 nanomaterials-14-00908-f010:**
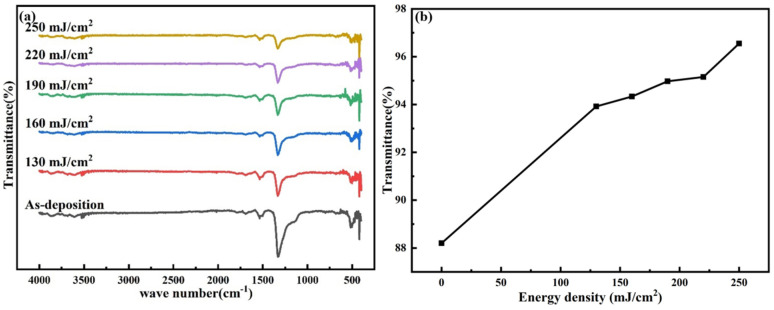
FTIR test results of Tb:In_2_O_3_ thin films treated with different energy densities: (**a**) FTIR test curve, (**b**) absorption peak (in 500 nm) transmittance.

**Figure 11 nanomaterials-14-00908-f011:**
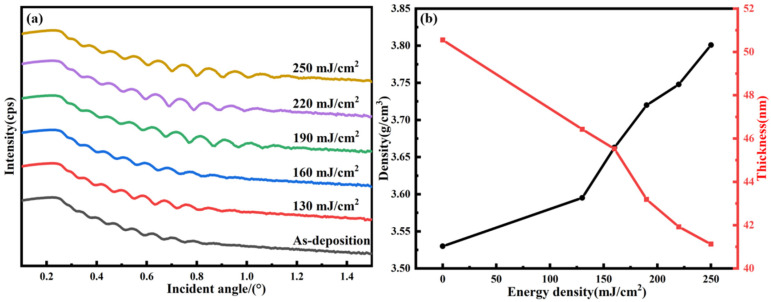
XRR test results of Tb:In_2_O_3_ thin films treated with different energy densities: (**a**) XRR test curve, (**b**) XRR fitting results.

**Figure 12 nanomaterials-14-00908-f012:**
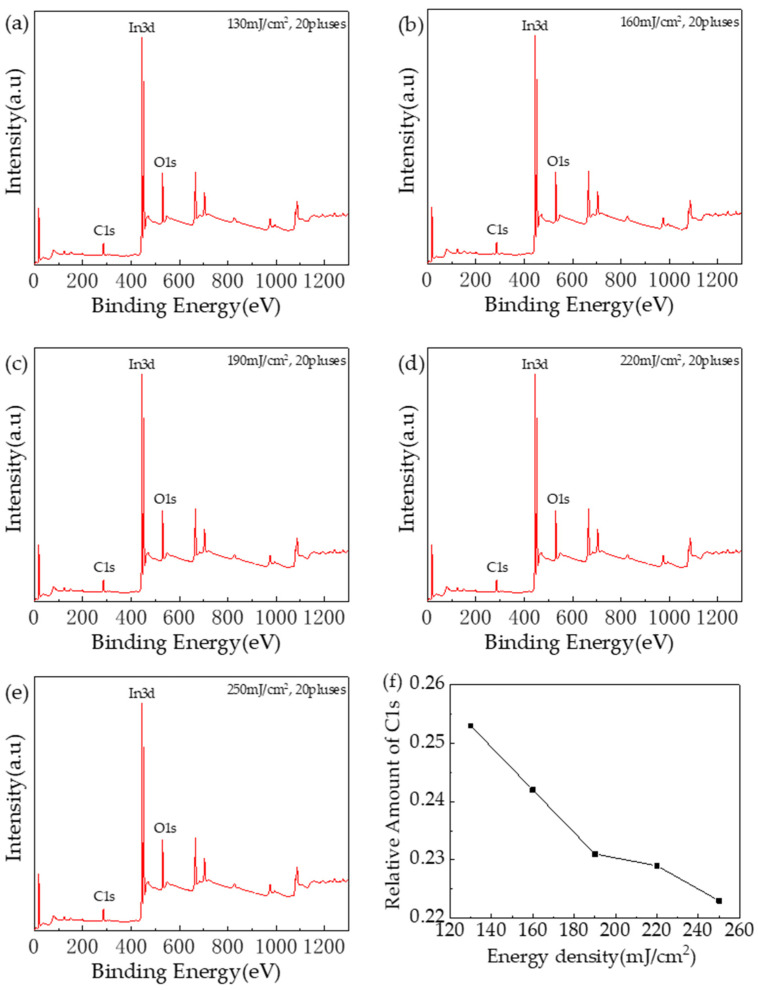
XPS testing of Tb:In_2_O_3_ thin films: (**a**–**e**) XPS full spectrum; (**f**) percentage of area occupied by C1s peak.

**Figure 13 nanomaterials-14-00908-f013:**
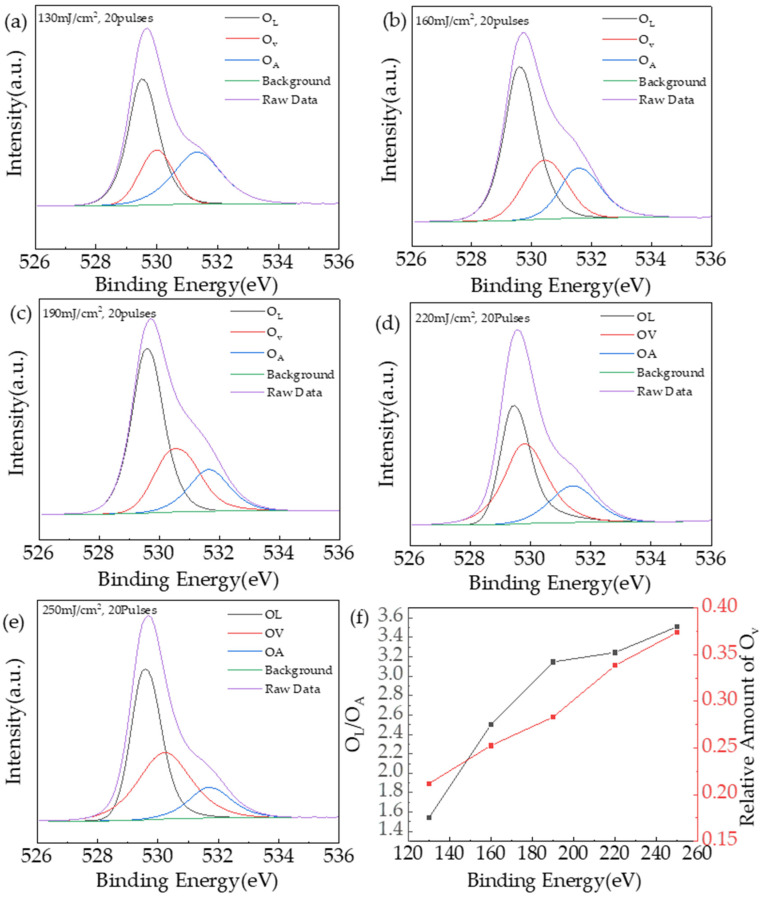
O1s fine spectrum test of Tb:In_2_O_3_ thin film: (**a**–**e**) O1s peak and fitting results; (**f**) the ratio of relative content of O_L_ and O_A_ (black line) and the relative content of O_V_ (red line).

**Figure 14 nanomaterials-14-00908-f014:**
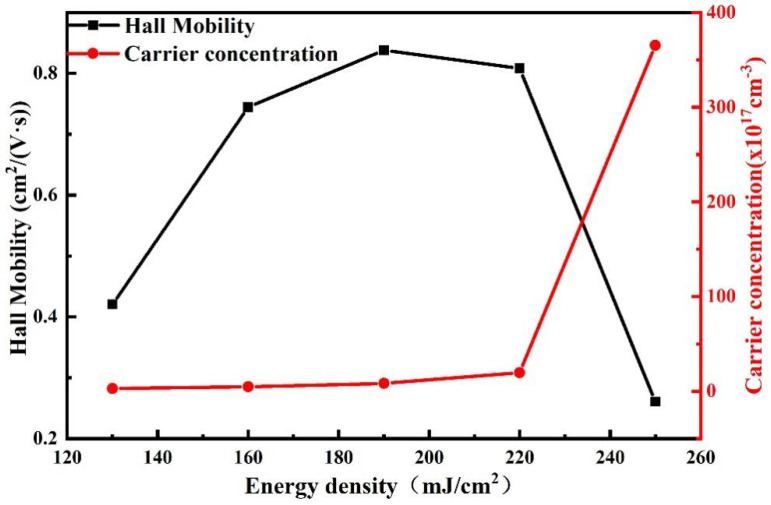
Electrical characteristic curves of Tb:In_2_O_3_ thin films treated with different energy densities.

**Figure 15 nanomaterials-14-00908-f015:**
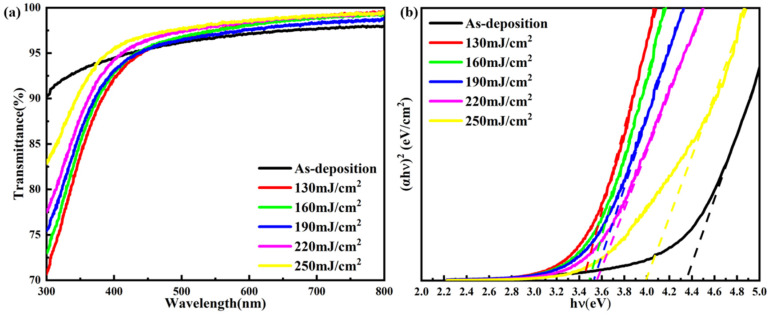
UV–Vis test results of Tb:In_2_O_3_ thin film treated with different laser energy densities: (**a**) transmittance curve; (**b**) optical bandgap fitting results.

**Figure 16 nanomaterials-14-00908-f016:**
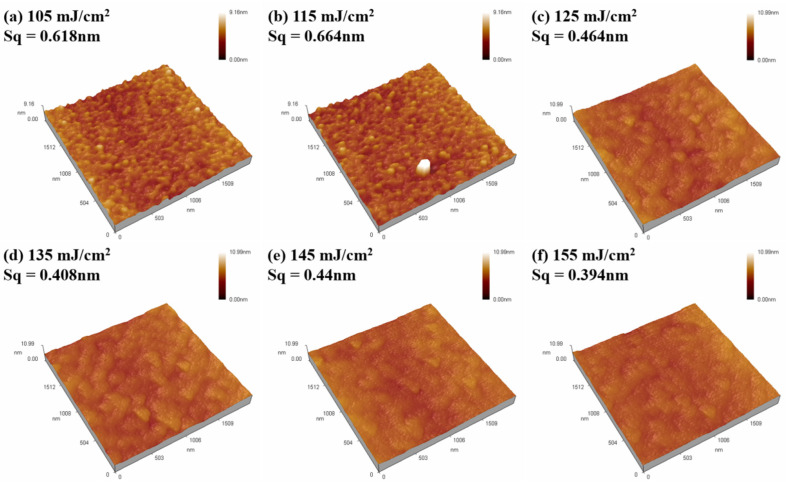
AFM images of Tb:In_2_O_3_ active layer thin films. Different energy densities: (**a**) 105 mJ/cm^2^; (**b**) 115 mJ/cm^2^; (**c**) 125 mJ/cm^2^; (**d**) 135 mJ/cm^2^; (**e**) 145 mJ/cm^2^ and (**f**) 155 mJ/cm^2^.

**Figure 17 nanomaterials-14-00908-f017:**
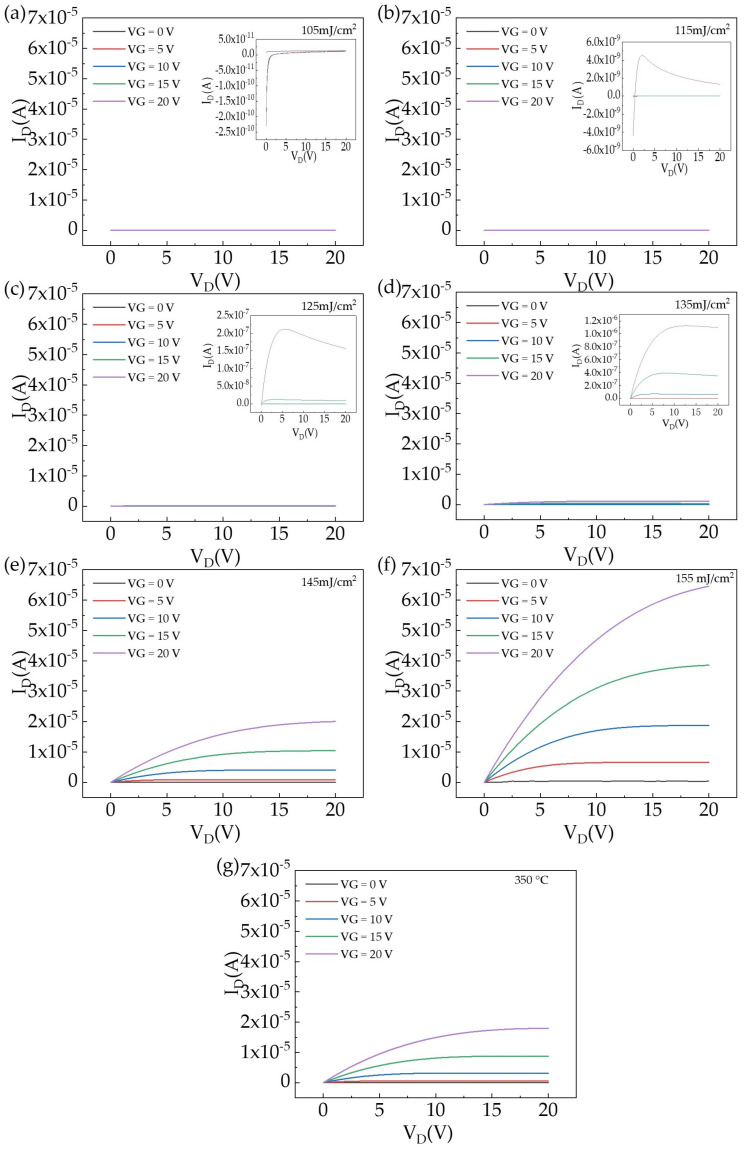
Tb:In_2_O_3_-TFT output curve: (**a**–**f**) different energy densities; (**g**) 350 °C heat treatment.

**Figure 18 nanomaterials-14-00908-f018:**
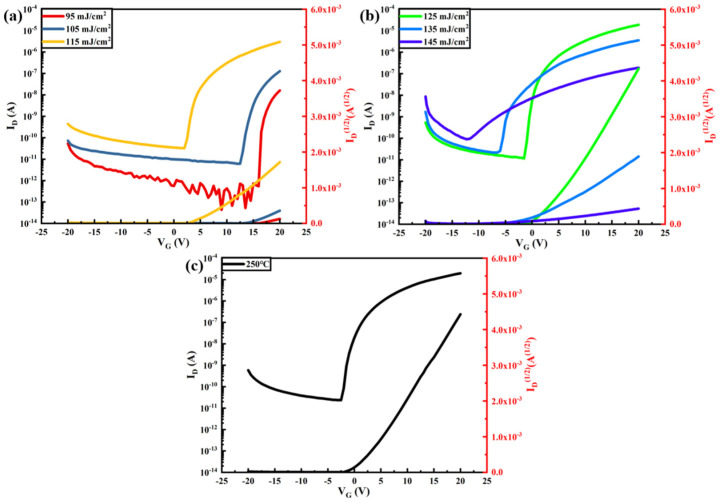
In_2_O_3_-TFT transfer curve: (**a**,**b**) different energy densities; (**c**) 350 °C heat treatment.

**Figure 19 nanomaterials-14-00908-f019:**
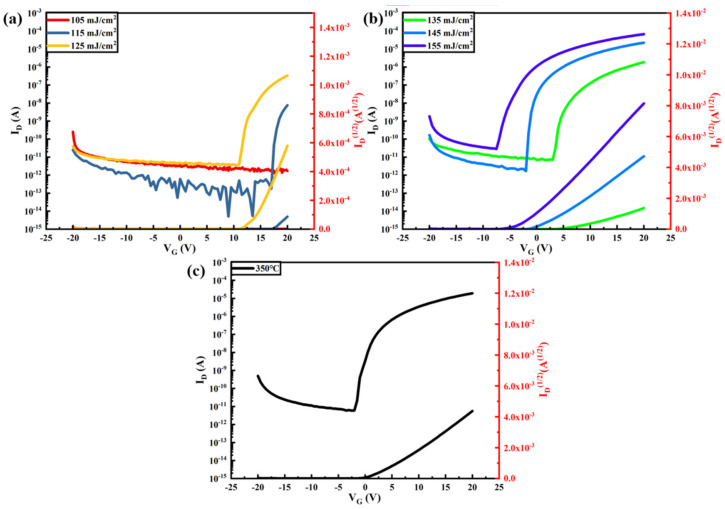
Tb:In_2_O_3_-TFT transfer curves: (**a**,**b**) different energy densities; (**c**) 350 °C heat treatment.

**Figure 20 nanomaterials-14-00908-f020:**
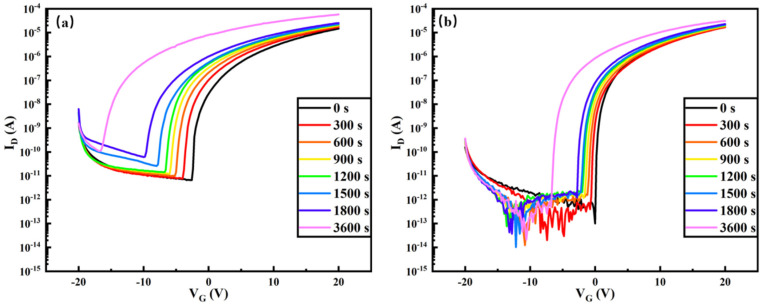
NBIS stability test: (**a**) In_2_O_3_-TFT; (**b**) Tb:In_2_O_3_-TFT.

**Figure 21 nanomaterials-14-00908-f021:**
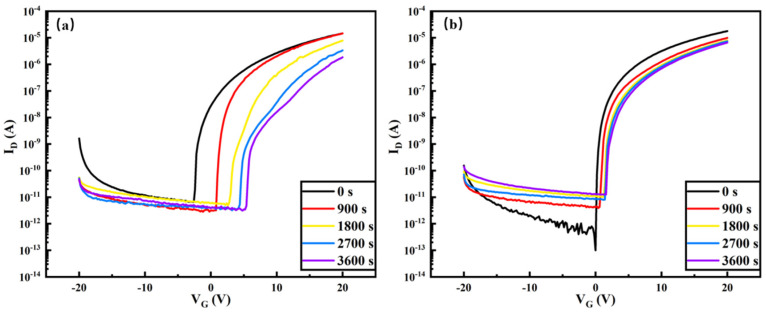
PBIS stability test: (**a**) In_2_O_3_-TFT; (**b**) Tb:In_2_O_3_-TFT.

**Table 1 nanomaterials-14-00908-t001:** Performance parameters of In_2_O_3_-TFT devices (subthreshold swing (SS), and threshold voltage (V_th_)).

Energy Density (mJ/cm^2^)	Mobility (cm^2^/V·s)	I_on_ (A)	I_off_ (A)	I_on_/I_off_	SS (V/decade)	V_th_ (V)
95	0.03	1.61 × 10^−8^	4.33 × 10^−14^	3.72 × 10^5^	0.20	16.15
105	0.08	1.28 × 10^−7^	6.09 × 10^−12^	2.10 × 10^4^	0.63	14.21
115	0.29	2.95 × 10^−6^	3.28 × 10^−11^	8.99 × 10^4^	0.60	4.17
125	1.31	1.88 × 10^−5^	1.16 × 10^−11^	1.63 × 10^6^	0.43	2.71
135	0.17	3.57 × 10^−6^	2.15 × 10^−11^	1.66 × 10^5^	0.62	−0.57
145	0.004	1.89 × 10^−7^	8.97 × 10^−11^	2.11 × 10^3^	4.32	−3.75
Heat Treatment	1.26	1.96 × 10^−5^	2.34 × 10^−11^	8.37 × 10^5^	0.49	0.67

**Table 2 nanomaterials-14-00908-t002:** Performance parameters of Tb:In_2_O_3_-TFT devices.

Energy Density (mJ/cm^2^)	Mobility (cm^2^/V·s)	I_on_ (A)	I_off_ (A)	I_on_/I_off_	SS (V/decade)	V_th_ (V)
105	×	×	×	×	×	×
115	0.0221	7.53 × 10^−9^	5.12 × 10^−15^	1.47 × 10^6^	0.56	17.35
125	0.111	3.35 × 10^−7^	3.43 × 10^−12^	9.76 × 10^4^	0.50	13.251
135	0.159	1.85 × 10^−6^	6.92 × 10^−12^	2.68 × 10^5^	0.46	6.18
145	1.07	2.23 × 10^−5^	1.66 × 10^−12^	1.34 × 10^7^	0.43	1.21
155	2.16	6.62 × 10^−5^	2.85 × 10^−11^	2.33 × 10^6^	0.76	−2.516
Heat Treatment	0.94	1.91 × 10^−5^	5.84 × 10^−12^	3.27 × 10^6^	0.39	2.12

## Data Availability

Data are contained within the article.

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
