# Peer review of "Effects of Laser Treatment of Terbium-Doped Indium Oxide Thin Films and Transistors"

_nanomaterials, 2024, doi:10.3390/nano14110908_

Round 1
Reviewer 1 Report
Comments and Suggestions for Authors
This paper is dedicated to investigation of the effects of laser treatment of terbium-doped indium oxide thin films and transistors which were fabricated by solution methods. Structural and electric properties were studied and discussed at a comparative basis.
The subject is currently of interest and such a study is timely and presents a valuable research direction. It also matches well the profile of Nanomaterials journal.
The results are also presented in enough detail. Characterization (XPS) and electric measurement techniques (and experimental details) are adequate, necessary and in fact routinely applied and well presented.
This manuscript clicks most boxes for novelty and scientific rigor so is acceptable for publication in Nanomaterials after minor revision mostly down to improvements of text and context:
1: Title can be shortened; it is too descriptive. Also, it should not begin with “Study on …” which is a banality but with “Effects of…”
2: The authors mention in many phrases the general designation “precursor solution”, and “precursor solution of TbOx”. However, they should clearly explain what the physical nature of these precursor species is, their abundance in the solution and their role in the process.
3: Fig. 17, 18, 19, 21, and also 13, 12, 10, most particularly Fig. 6: Legends, and axes marks are extremely small and difficult for reading. All figures should have the same style and similar readability.
4: Introduction lacks scope and broadness, most essentially that in the case of Indium oxides, theoretical publications exist (e.g., CrystEngComm 23 (2021) 6661-6667) that show structural understanding and guidance to synthesized thin films and low-dimensional material systems, with a plethora of implications for metal oxides structure and electronic properties.
5: The discussion of structural and compositional homogeneity of samples is a bit scattered within the Discussion section, maybe it is worth to separately summarize homogeneity issues in a special paragraph.
6: The conclusions should appear in a separate section with the corresponding subtitle. Conclusions can be improved further by underlining even more the practical value of why these results are of importance for concrete applications application in large-scale device arrays so to improve even more the general message of this paper.
7: The authors could consider submitting some of the figures as supplementary material.
Comments on the Quality of English LanguageStylistic and grammatical revision of the text would be beneficial.
Author Response
Dear Editors:
On behalf of my co-authors, we thank you very much for giving us an opportunity to revise our manuscript, we appreciate editor and reviewers very much for their positive and constructive comments and suggestions on our manuscript entitled”Effects of laser treatment of terbium-doped indium oxide thin films and transistors“.(nanomaterials-2980222)
We have studied reviewer’s comments carefully and have made revision which marked in red in the paper. We have tried our best to revise our manuscript according to the comments. Attached please find the revised version, which we would like to submit for your kind consideration.
We would like to express our great appreciation to you and reviewers for comments on our paper. Looking forward to hearing from you.
Thank you and best regards.
Yours sincerely,
Dingrong Liu
Corresponding author:
Name: Honglong Ning
E-mail: ninghl@scut.edu.cn

Reviewer 2 Report
Comments and Suggestions for Authors
The manuscript, titled “Study on laser treatment of terbium-doped indium oxide thin films and transistors prepared by solution methods”, submitted to MDPI Nanomaterials by Yao et al. is an experimental study on the fabrication of terbium-doped Tb:Inâ‚‚O₃ thin films, prepared by sol-gel deposition and laser-assisted sintering, for applications as a active semiconductor layer in metal-oxide-based field-effect thin film transistors.
The authors have successfully synthesised sol-gel solution, based on Tb- and In- nitrate precursors in ethylene glycol methyl ether solvent, which was then spin coated onto a pre-fabricated SiO2/Al:Nd/AlOx:Nd multilayer to serve as a substrate/gate electrode/insulating layer, dried and laser-treated the Tb:Inâ‚‚O₃ sol-gel layer with KrF excimer laser at varying power density and performed a thorough characterization of the resulting layers by means of: XPS to amass information on the chemical state of the terbium dopant and oxygen vacancies as a function of laser treatment dose; UV-Vis spectroscopy to determine optical band-gaps; AFM to provide details about surface roughness; FTIR to trace the decrease in residual organics from the sol-gel; XRD to confirm that no-recrystalization is observed, including reflectance measurements for thickness determination. Detailed electrical measurements were also obtained, including the Hall mobility and carrier concentration, and TFT transfer and output curves, and positive- and negative bias illumination stress (PBIS, NBIS) measurements. Some of the Tb:Inâ‚‚O₃ is also compared to reference pure TbOx (for chemical characterization) and Inâ‚‚O₃ (for electrical measurements) samples and devices. The authors show that the laser-assisted sol-gel treatment yields thin films more applicable for TFT active layer applications, compared to thermal treatment, and that (in this case) the optimal treatment dose for higher performance was found to be 145 mJ/cm².
Overall, I am impressed by the sheer amount of experimental work conducted for the manuscript. The UV processing, and especially laser assisted processing, of sol-gel layers is a very active development in the field of wet-chemical semiconductor device fabrication, so the work has novelty and would be of interest for the readers of Nanomaterials. The quality of the English presentation, while stylistically not on a native-speaker level, is good enough for the text to be readable.
I would be in favour for the acceptance of the manuscript, however, there are some major issues with its presentation, a compete lack of structure, as well as plenty of important details lacking and some very confusing inconsistencies across the text. I believe, that in many cases this would be enough for a paper to be desk-rejected, however, given that I am impressed by the amount of experimental work; and additionally - can spot that there is a clear research idea being presented, I would support that this work is considered for a publication, after some major improvements.
Here are the suggestions:
(1) Starting with the abstract (which is pretty much the only logically separate entity in the text, at this point), there are way too many abbreviations, that are not introduced in it (MOTFT; IGZO(-TFT); and most importantly NBIS and PBIS, which are not introduced at all at any moment in the entire length of the text.
(2) The entire rest of the text is laid in an unstructured manner, as if the manuscript was planned to be in a Letter style. I strongly suggest that the text is re-structured in a proper Introduction; Materials and Methods; Results and Discussion; Conclusions format, since it is difficult to follow.
(3) The introductory part (lines 29-85) seems acceptable (apart from the lack of clarity what is “high coefficient in metal oxide thin films” - line 80 - absorption coefficient ?).
The materials and methods (lines 86-147), however has issues:
-> I suggest that proper chemical notations, as well as information about the manufacturer/source of the materials used is added.
-> this section mentions the formulation of two sol-gel solutions (Tb:In₂O₃ and TbOx). Later in the text (starting line 399 and Fig. 18) suddenly some mentions about pure In₂O₃ sample is also mentioned. It should be added if the same formulation, excl. the Tb-precursor is used.
-> The text between lines 106-113 and then 124-130 is absolutely identical. Was this a repetition ?
-> More details about the sputtering and processing (etching / anodisation) of Al:Nd layer should also be mentioned, or if the same approach was used in the author’s previous works could be referred to instead.
-> The geometry of the TFT is almost non-descript. I suggest that either a top-down drawing is added to Figure 1 to show the gate geometry and the position of the source/drain electrodes or that it is explained in the text (i.e., no dimensions of the T-shaped gate are given, no information about the top-electrodes distance, even no mention is made that Al was deposited (nor how) to form them - in fact they are not even mentioned in this section).
-> It is not mentioned what was the purpose of the UV-patterning (and what was the pattern - it is mentioned before the repetition of the spin-coating process, where the Tb:In₂O₃ sol was added so it is not clear to me, as a reader whether: (a) it is also a repetition of the UV treatment of the substrate; (b) was the sol-layer patterned, or the AlOx:Nd layer on the substrate; (c) was this another way to mention the laser treatment in advance; (d) what is it actually improving in the Tb:In₂O₃ layer.
-> All throughout the text the authors relate some properties to energy densities. It is a bit confusing to me why this value is used (if this is the energy density of the laser), when 20 pulses are applied (i.e. does it imply 20 pulses of X mJ/cm², yielding a cumulative dose of 20 x X mJ/cm² or the authors simply prefer to refer only to the laser output, and not the total amount of energy received by the sol layer, and needed to cure it.
-> In line 133 - the energy densities listed are 130, 160, 190, 220, and 250 mJ/cm² and then again in line 135 another set of numbers is listed: this time 105, 115, 125, 135, 145, and 155 mJ/cm². I don’t really understand, why different UV dose sets were prepared for characterization and electrical measurements (I could assume, that given the results from characterisation the authors decided to do the measurements only in a samples treated in a narrower dose set). Nevertheless, it is confusing - I suggest that at least a clarification is added to help the reader understand that the two sets were used for different purposes (and I know that this is implied by “films” in line 133, and “-TFT devices” in line 135, but it is not immediately clear).
(4) In the thin-film characterisation part of the results (lines 148-380) I also have some issues:
- It feels as the authors tried to include as much characterisation measurements as possible, but some of these figures really could be moved in a supporting information file, since they bulk up the figure numbers w/o contributing too much to the discussion, alternatively they could be altered or better implemented in the text:
* (Figure 2) I understand that the point of this figure is to show the match between the laser excitation wavelength and the sol, but probably this could be moved to the supporting information - also a line could be added to indicate the 248 nm KrF position, since the scaling (200 - 800 nm) is huge for clarity; Additionally, the correct term is “Absorbance (a.u.)” and not absorption.
* (Figure 4). I do not really understand the purpose of this spectra - XPS data is sufficient to prove the Tb3+/Tb4+ existence in the samples; also the two reference spectra (from ref. 36) seem very suspicious- the Tb3+ seems as if it is in a complexed state, and additionally I checked ref. 36 and does not seem to include any such optical data so I am wondering whether the ref. numbering is correct and where this data is taken from ?
* (Figure 6) Obviously the PL spectra is dominated by the substrate, but this analysis is way too uninformative to make any conclusions about the relaxation mode of the Tb dopant. May the authors consider simply that the layers (later revealed to be < 50 nm thick) are not observable in their PL measurements due to lack of sensitivity.
* (Figure 7) I don’t understand, and it is not clarified in the text, is this figure simply a graphical representation of the theory described in lines 219-225 (if so, some reference to literature could be made); or a result from actual measurements?
* (Figure 8) A reference to the source of the reflectance peaks positions should be made (PDF number or ICDB number, etc.) also the Miller indices are more correctly noted without comas (e.g. (222), instead of (2,2,2)). Also - given that the layers are extremely thin - was grazing incidence used here ?
* (Figure 10b) It is stated “absorption peak transmittance” I can prevent myself from commenting on the expression itself, but it should be added - which peak was actually used to make this data?
* (Figure 12) The survey XPS scans could be omitted or moved to the supporting information. The scaling makes it impossible to visually see any changes in the C1s peak intensity.
- Line 356. “Tacu’s formula” probably refers to the Tauc equation.
- Additionally, throughout this part of the text there are a lot of typos, associated with misuse of subscripts and superscripts: e.g. (mJ/cm2 vs. mJ/cm²) and (In2O3 vs Inâ‚‚O₃)
(5) In the electrical characterisation of TFT devices part (Lines 381 - 494) I can only notice some issues:
- In this part, suddenly the reader is brought to the realisation that apart from the laser-treated TFT devices, there is also a reference undated Inâ‚‚O₃ one (the preparation of which is not mentioned in the materials section); and also that there are measurements conducted on heat-treated TFT devices (albeit it is not clear at which temperature, since around Fig. 17 it is claimed 350 °C, but then changes to 250 °C in Figure. 18). The authors should revise - (a) including details about the preparation and reasoning for these reference samples in an appropriate section; and (b) improve the consistency.
- Also all abbreviations should be explained properly and timely PBIS, NBIS, etc. Tables captions should also include a legend of the notations used in each table (e.g. SS, Vth, etc.) to outline the meaning of the contents and be self-explanatory.
Author Response

(The authors gave the same response as above.)

Round 2
Reviewer 2 Report
Comments and Suggestions for Authors
I would like to thank the authors for addressing the comments, raised by the reviewers. I am a bit dissapointed that they did not provide any point-by-point response, but simply an updated version of the paper instead, but tracing the points, raised by my comments, I can see that they were implemented. I would support the publication of the manuscript, after the revisions now, but would urge the authors to follow a proper reviewer reply protocol upon next revision.
